# GURU: GUIDED UNLEARNING VIA RESIDUAL UNCERTAINTY IN REGRESSION MODELS

## ABSTRACT

Machine Unlearning (MU) addresses the fundamental requirement of removing the influence of specific data samples from trained models, a need that arises in privacy-sensitive applications where individuals can request the deletion of their personal information in accordance with the "right to be forgotten" principle. While unlearning has been actively studied in classification domains (computer vision, language modeling, speech), its extension to regression remains underexplored both methodologically and theoretically. In this work, we redefine unlearning and its evaluation in regression by building on recent advances in the broader field of MU. We propose GURU (Guided Unlearning via Residual Uncertainty), the first distillation-based method for regression unlearning. GURU derives predictive uncertainty directly from residual errors by scaling residuals with a confidence factor, thereby defining a Gaussian predictive distribution that captures both the model's prediction and its confidence, providing a probabilistic view of regression outputs. This formulation enables a closed-form Kullback–Leibler divergence objective between teacher models and the student. We validate GURU on four regression datasets spanning vision and language. We show that GURU achieves competitive performance in terms of efficacy (removal of target information), utility (preservation of retained knowledge), and efficiency (computational cost). In addition, we propose GRUM, a regression-aware extension of the Global Unlearning Metric (GUM) that jointly considers all previous principles.

## 1 INTRODUCTION

Recent advances in privacy-preserving machine learning have drawn notable interest in *Machine Unlearning* (MU), the task of removing the influence of specific training data from a learned model. This capability has become increasingly important with the introduction of numerous data protection regulations such as the EU's General Data Protection Regulation (GDPR) (Voigt & Von dem Bussche, 2017), which formally grants users the "right to be forgotten" (Mantelero, 2013). The most straightforward way to achieve this goal would be to retrain the model from scratch after removing all data corresponding to individuals who request deletion. However, such a strategy is typically impractical due to the prohibitive computational, environmental, and financial costs of training modern large-scale models (Crawford, 2022).

While significant progress has been made in classification settings, extending unlearning to regression tasks remains largely unexplored. In recent years, a wide range of works have been made for classification-based unlearning applied across a variety of domains, including computer vision (Foster et al., 2024; Fan et al., 2023), natural language processing with large language models (LLMs) (Maini et al., 2024; Liu et al., 2024), and speech processing (Koudounas et al., 2025; Choi et al., 2025). Despite these advances, all these works rely on the discrete nature of classification problems, leveraging characteristics such as the distribution of output logits (Kurmanji et al., 2023) or the discrete and finite nature of classification (Chen et al., 2023). Regression problems, on the other hand, are generally characterized by continuous, single-valued outputs. As a consequence, regression unlearning presents distinct theoretical and practical challenges that remain insufficiently addressed in the literature. To the best of our knowledge, only Tarun et al. (2023) has attempted to address this gap by proposing the first dedicated methods (Blindspot Unlearning and Gaussian Amnesiac Learning) and metrics for general deep regression unlearning.

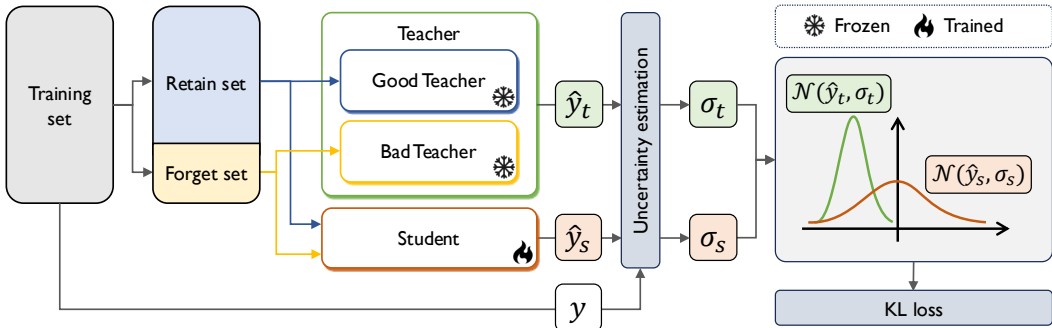

Figure 1: Overview of the GURU unlearning pipeline. The good teacher (frozen original model) guides the student on $\mathcal{D}_r$, while the bad teacher (input-agnostic sampler) defines targets on $\mathcal{D}_f$. Both teachers and the student produce residual-based Gaussian predictions $(\mu, \sigma)$, and the student is optimized by minimizing the KL divergence between its distribution and the appropriate teacher.

To address these shortcomings, we introduce GURU (Guided Unlearning via Residual Uncertainty), the first distillation-based approach for regression unlearning. In GURU, regression outputs are recast as probabilistic predictions by deriving predictive uncertainty directly from residual errors. Residuals are scaled into a variance term, yielding a Gaussian distribution for each prediction that jointly encodes its mean and confidence. Unlearning is then formulated as a teacher-student distillation problem, where the student model minimizes a closed-form Kullback-Leibler divergence between its Gaussian predictive distribution and those provided by the two teachers.

On the evaluation side, we build on the Generalized Unlearning Metric (GUM) introduced by Koudounas et al. (2025) and propose GRUM, a regression-aware extension that jointly captures the three fundamental desiderata of unlearning: *efficacy* (removal of target information), *utility* (preservation of useful knowledge), and *efficiency* (computational cost) (Hayes et al., 2024).

We validate GURU across four regression datasets spanning vision and language, all of which contain information at the individual level, thereby emulating realistic unlearning scenarios where data deletion is legally mandated. Experimental results show that GURU establishes a new state-of-the-art for regression unlearning, outperforming previously proposed regression methods.

In summary, our main contributions are as follows:

- We introduce GURU, the first regression unlearning method based on teacher–student distillation. GURU derives predictive uncertainty from residuals and defines a Gaussian distribution over outputs, enabling a closed KL-divergence objective.

- We propose GRUM, a regression-aware evaluation metric that extends GUM to evaluate all crucial aspects of an unlearning method jointly.

- We conduct extensive experiments on four vision and language regression datasets containing individual-level information, demonstrating that GURU achieves competitive unlearning performance in realistic deletion scenarios.

## 2 BACKGROUND

**Machine Unlearning.** Machine Unlearning refers to the process of removing the influence of specific training samples from a machine learning model. Formally, let a training set $\mathcal{D}$ be partitioned as $\mathcal{D} = \mathcal{D}_r \cup \mathcal{D}_f$, and $\mathcal{D}_r \cap \mathcal{D}_f = \emptyset$, where $\mathcal{D}_r$ is the *retain set* (data that should be preserved) and $\mathcal{D}_f$ is the *forget set* (data whose influence is to be removed). In addition, we assume access to a disjoint *test set* $\mathcal{D}_t$, never used during training or unlearning, to assess the generalization capabilities of a model. Let $\mathcal{T}(\cdot)$ denote the training algorithm applied to a dataset, and $\mathcal{U}(\cdot, \mathcal{D}_f, \mathcal{D}_r)$ represent an unlearning process applied to a model. Within MU, three reference models are defined:

$$M^{(o)} = \mathcal{T}(\mathcal{D}), \qquad M^{(g)} = \mathcal{T}(\mathcal{D}_r), \qquad M^{(u)} = \mathcal{U}(M^{(o)}, \mathcal{D}_f, \mathcal{D}_r),$$

where $M^{(o)}$ is the *original model* trained on the full dataset $\mathcal{D}$ before any deletion request, $M^{(g)}$ is the *gold model* obtained by training from scratch on the retain set $\mathcal{D}_r$ only, and $M^{(u)}$ is the *unlearned model* produced by an unlearning algorithm $\mathcal{U}$ applied to $M^{(o)}$. If $M^{(u)} \equiv M^{(g)}$, we refer to $\mathcal{U}$ as an *exact unlearning* process. Exact unlearning is only applicable to limited cases. In general, *approximate unlearning* is considered to be achieved if $M^{(u)} \approx M^{(g)}$.

**Distillation-Based Unlearning.** A prominent approach in the unlearning literature is the *teacher-student distillation* framework (Chundawat et al., 2023). In this case, unlearning is formulated as a re-teaching process: a student model is trained to imitate a *good teacher* – a model trained on the full dataset – for samples belonging to the retain set $\mathcal{D}_r$, while concurrently matching a *bad teacher* on samples belonging to the forget set $\mathcal{D}_f$. The bad teacher typically corresponds to a randomly initialized network, $M^{(rnd)}$.

The unlearning process is obtained by minimizing the following loss function:

$$\mathcal{L} = \sum_{x \in \mathcal{D}_f} \mathrm{KL}(M^{(rnd)}(x) \,\|\, S(x)) + \sum_{x \in \mathcal{D}_r} \mathrm{KL}(M^{(o)}(x) \,\|\, S(x)) \tag{1}$$

The central mechanism involves minimizing a KL divergence loss: the student is encouraged to match the output distribution of the good teacher on $\mathcal{D}_r$ and align with the bad teacher on $\mathcal{D}_f$. This contrastive distillation approach fundamentally relies on access to meaningful output distributions – typically logits – making it naturally suited to classification tasks with discrete label spaces. For this reason, extending teacher-based unlearning to regression settings introduces conceptual challenges. In continuous-output models, predictions are scalar and do not naturally define a probability distribution over classes.

**Unlearning Evaluation Principles.** A meaningful evaluation of unlearning methods must consider three complementary principles: *efficacy*, *utility*, and *efficiency* (Hayes et al., 2024).

*Efficacy* measures the degree to which $\mathcal{D}_f$ has been erased from the unlearned model $M^{(u)}$. In other words, the unlearned model $M^{(u)}$ is expected to produce a behavior similar to that of the gold model $M^{(g)}$, on *forget* samples. The most widely used proxy for efficacy is the membership inference attack (MIA) (Chen et al., 2021; Graves et al., 2021). In the context of MU, the MIA score is the accuracy obtained by an adversary $\mathcal{A}_{\mathrm{MIA}}$, tasked with solving a binary problem: given access to the unlearned model's outputs over a set of samples, the adversary should distinguish whether a sample $x$ belongs to the forget set $\mathcal{D}_f$ or to a test set $\mathcal{D}_t$ ($\mathcal{A}_{\mathrm{MIA}}(M(x)) \mapsto \{f, t\}$). If unlearning is successful, the behavior of $M^{(u)}$ over samples from $\mathcal{D}_f$ should be indistinguishable from the behavior over samples from $\mathcal{D}_t$. In practice, the literature usually aligns the MIA score of the unlearned model with that of the gold model $M^{(g)}$, which by definition has never been exposed to the forget set.

*Utility* refers to the ability of the unlearned model $M^{(u)}$ to preserve the knowledge of $\mathcal{D}_r$ acquired during training, and to maintain generalization properties on $\mathcal{D}_t$. In other words, while the influence of the forget set should be removed, the model should still perform well on all data that is meant to be retained. A central challenge here is to avoid *catastrophic forgetting*, the phenomenon that occurs when the process of unlearning causes the model not only to forget the targeted data $\mathcal{D}_f$, but also to inadvertently lose knowledge about $\mathcal{D}_r$ (Jagielski et al., 2022).

*Efficiency* concerns the cost of unlearning relative to full retraining. The most straightforward way to satisfy a deletion request, in fact, would be to discard the original model and train a new one from scratch on $\mathcal{D}_r$. However, this procedure is typically infeasible in practice. MU methods are expected to achieve results comparable to the gold model, but at a fraction of the cost of full retraining. Efficiency is thus a crucial principle: an approach that provides high efficacy and utility but demands nearly the same resources as full retraining defeats the practical purpose of unlearning.

Together, these three principles form the foundation of unlearning evaluation: a method should forget effectively (*efficacy*), preserve useful knowledge (*utility*), and do so at a reduced cost (*efficiency*).

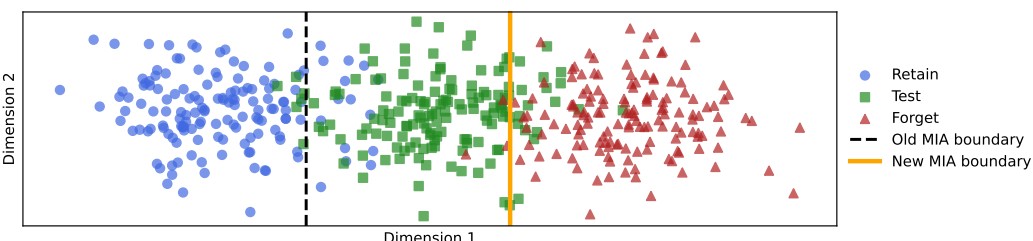

Figure 2: Toy example illustrating a possible MIA evaluation after unlearning. Forget samples (red) remain separable from test samples despite being classified as test by the old MIA boundary (dashed line). The new MIA boundary (solid line) correctly captures this separation.

## 3 RELATED WORKS

**MU in regression tasks.** Compared to classification, MU in regression tasks remains largely underexplored. To the best of our knowledge, the only general study to date, Tarun et al. (2023), introduced two methods: *Blindspot Unlearning*, which leverages a partially trained model to guide the forgetting process, and *Gaussian Amnesiac Learning*, which randomizes targets sampled from a Gaussian distribution intended to mimic the behavior of the full training set. Within that study, the authors used two baseline unlearning methods borrowed from classification to test their methods: *Finetuning*, where the model is updated only on the retain set to mitigate the effect of the forget set, and *NegGrad*, which applies stochastic gradient ascent to the forget set to erase its influence. However, the evaluation protocol used in that work presents structural limitations when considered in light of recent advances in unlearning. Beyond this work, domain-specific studies in areas such as load forecasting (Xu & Teng, 2024) and recommender systems (Liu et al., 2022) have highlighted the importance of unlearning for regression tasks. These efforts, although shaped for particular applications, underscore the need for systematic and generalizable approaches to regression unlearning.

**Evaluation in regression unlearning.** In classification, MIA typically exploits the distribution of output logits (Hayes et al., 2024). Regression models, however, only produce a single continuous value; therefore, this strategy is not very informative. To overcome this, Tarun et al. (2023) feed richer features into a support vector classifier, including the model's prediction, penultimate-layer gradients, and hidden activations. By augmenting the input in this way, they provide the attacker with sufficient information to compensate for the lack of logits, making the attack effective in regression. Their evaluation, however, has a fundamental limitation. They train the attacker to distinguish between $\mathcal{D}_r$ and $\mathcal{D}_t$ and then apply it to $\mathcal{D}_f$, assuming that predicting it as "test-like" proves successful unlearning. In practice, this setup never directly compares forget samples to the actual test distribution. Instead, it only ensures that $\mathcal{D}_f$ is not similar to $\mathcal{D}_r$, which can mask cases of *overunlearning* (Shi et al., 2024). In such cases, the model may superficially label forget samples as test while still clustering them far from the actual $\mathcal{D}_t$. Figure 2 shows exactly this failure: the dashed MIA boundary misclassifies almost all $\mathcal{D}_f$ as test, although they remain separable from $\mathcal{D}_t$.

**GUM.** While a wide range of works report results on isolated aspects of the three dimensions of unlearning, the only unified metric that jointly considers all three is the Global Unlearning Metric (GUM) (Koudounas et al., 2025). First, a normalized score is defined for each desideratum:

$$E = 1 - \left( \frac{\min\{\omega^{(u)}, \omega^{(o)}\} - \min\{\omega^{(g)}, \frac{\omega^{(u)} + \omega^{(o)}}{2}\}}{\omega^{(o)} - \min\{\omega^{(g)}, \frac{\omega^{(u)} + \omega^{(o)}}{2}\}} \right)^2 \tag{2}$$

$$U = 1 - \left| \psi^{(g)} - \psi^{(u)} \right|, \qquad T = 1 - \frac{\log\left(\theta^{(u)} + 1\right)}{\log\left(\theta^{(g)} + 1\right)} \tag{3}$$

Here, $\omega$, $\psi$, and $\theta$ denote generic measures of *efficacy* ($E$), *utility* ($U$), and *efficiency* ($T$), respectively. The superscripts specify the model variant used to compute the score: ($o$) refers to the *original model*, ($g$) to the *gold model*, and ($u$) to the *unlearned model*. The formulation is intentionally

abstract: in principle, any appropriate task-specific score can be used to instantiate these quantities. In the original GUM paper for classification, $\psi$ was instantiated as the macro-F1 score on the test set, $\omega$ as the MIA accuracy between $\mathcal{D}_f$ and $\mathcal{D}_t$, and $\theta$ as the runtime of the method. Finally, the three components are aggregated into a single measure through a weighted harmonic mean:

$$\text{GUM} \;=\; \frac{(1 + \alpha + \beta)\,UET}{\alpha ET + \beta UT + UE} \tag{4}$$

where $\alpha$ and $\beta$ are hyperparameters, commonly set to 1 to balance all three dimensions equally.

## 4 METHODOLOGY

In this section, we introduce our two main contributions: the proposed unlearning method, GURU (Guided Unlearning via Residual Uncertainty), and the evaluation framework with the unified evaluation metric, GRUM (Global Regression Unlearning Metric).

### 4.1 GUIDED UNLEARNING VIA RESIDUAL UNCERTAINTY

One of the main challenges in adapting teacher-student unlearning to regression lies in the absence of a natural probability distribution over outputs, unlike in classification, where logits define such a distribution. To overcome this problem, we design GURU around a residual-driven estimation of the uncertainty of the model, enabling a closed-form KL divergence objective.

**Teachers, student, and target definition.** We employ a dual-teacher setup to guide the distillation process of the student model (i.e., the model undergoing unlearning). More specifically, we include (i) a *good teacher*, i.e. the frozen original model trained on the full dataset, providing predictions $M^{(o)}(x)$ for samples in the retain set; and (ii) a *bad teacher*, providing stochastic, input-agnostic predictions $\hat{y}_B \sim \mathcal{N}(y_r, \sigma_r^2)$, where $y_r$ and $\sigma_r$ summarize the target distribution of samples in the retain set $\mathcal{D}_r$). In other words, the good teacher provides accurate predictions for samples that need to be preserved, the bad teacher provides poor predictions for samples that need to be forgotten. Let $\xi(x) \in \{0, 1\}$ be an indicator function of whether a sample should be forgotten ($\xi{=}1$) or retained ($\xi{=}0$). The teacher's predicted mean value for a sample $x$ is defined as:

$$\hat{y}_t(x) \;=\; \xi(x)\,\hat{y}_B \;+\; \big(1 - \xi(x)\big)\,M^{(o)}(x). \tag{5}$$

Throughout the unlearning process, we define as $\hat{y}_s = M^{(u)}(x)$ the prediction of the student model, (i.e., the model undergoing unlearning).

**Residual-based uncertainty.** We make an assumption on the normality of the distributions of the student's and teachers' outcomes. Since the predicted values are indeed available, we use those as the means of the corresponding distributions. To estimate the uncertainty of the models' distributions, we use the residual between the prediction $\hat{y}$ (either from the student, or the teacher) and the ground truth value $y$. The corresponding standard deviation is obtained as:

$$\Sigma(y, \hat{y}) = \frac{|y - \hat{y}|}{z}, \tag{6}$$

where $z = \Phi^{-1}\big(\frac{1+c}{2}\big)$ corresponds to a desired confidence level $c$ (e.g., $c = 0.95$ yields $z \approx 1.96$). With this definition, the standard deviation obtained is the one that ensures that the ground truth $y$ lies at the boundary of a $c$-confidence interval around the prediction $\hat{y}$. This construction ensures that predictions closer to the ground truth are associated with small-variance Gaussian distributions, whereas inaccurate predictions have larger variance. Teacher and student standard deviations are estimated according to their respective residuals: $\sigma_t(x) = \Sigma(y, \hat{y}_t)$ and $\sigma_s(x) = \Sigma(y, \hat{y}_s)$.

Figure 3 illustrates this mechanism: for three different scenarios, the residual-driven construction produces Gaussian distributions of varying width around teacher and student predictions. The resulting KL divergence values (shown in the titles) quantify the discrepancy between the two predictive distributions. We note that we can only compute uncertainties in this way for training samples, for which the ground truth value is known. However, since the unlearning process only involves $\mathcal{D}_r$ and $\mathcal{D}_f$ data (which, together, constitute the training set), the assumption of availability of the targets holds by construction.

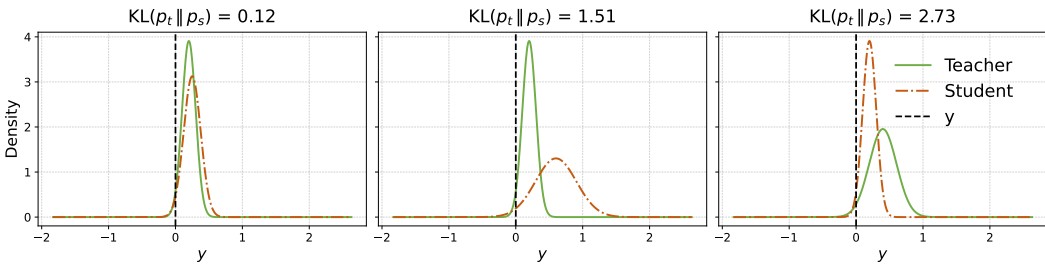

Figure 3: Residual-based uncertainty illustrated across three cases. Teacher (green) and student (orange) distributions are derived from their residuals with respect to the true target (black dashed line). The mean of each Gaussian corresponds to the model output $\bar{\mu}$, while $\bar{\sigma}$ is computed directly from the residual according to Eq. 6. The KL divergence values quantify the discrepancy between the predictive distributions, serving as the loss minimized during training.

**KL objective and training loss.** The distributions for the teacher, $p_t(y \mid x) = \mathcal{N}(\hat{y}_t, \sigma_t^2(x))$ and the student, $q_s(y \mid x) = \mathcal{N}(\hat{y}_s, \sigma_s^2(x))$, are now available. Based on the assumption of normality of the distributions, the Kullback–Leibler divergence $\mathrm{KL}(p_t \,\|\, p_s)$ can be computed in closed-form. With the goal of aligning the student with the teacher, the loss of interest can be defined (with dependencies on $x$, $y$ removed for ease of notation) as:

$$\mathcal{L}_{\mathrm{KL}}(x, y) \;=\; \log\frac{\sigma_s}{\sigma_t} \;+\; \frac{\sigma_t^2}{2\,\sigma_s^2} \;+\; \frac{\left(\hat{y}_t - \hat{y}_s\right)^2}{2\,\sigma_s^2} \;-\; \tfrac{1}{2}. \tag{7}$$

With the exception of the constant term $-\frac{1}{2}$ that can be ignored during the optimization process, this loss combines mean alignment and uncertainty calibration into a single closed-form objective. The quadratic term $(\hat{y}_t - \hat{y}_s)^2 / 2\sigma_s^2$ penalizes deviations in the means, with stronger penalties when the student is confident (small $\sigma_s$).

Figure 3 illustrates how this behavior supports the desiderata of unlearning. When teacher and student produce similar predictions and comparable uncertainties (left plot), the resulting Gaussians overlap, and the KL is close to zero, indicating that the student is already aligned. When the teacher is certain and the student not (center plot), the loss increases moderately and pulls the student closer to the teacher, improving accuracy while still tolerating uncertainty (*utility*). Finally, when the student is confident but the teacher is not (right plot, a typical case of forget samples), the quadratic penalty dominates and the KL is high; optimization pushes the student to follow the input-agnostic teacher distribution, thereby enforcing *efficacy*. Overall, the procedure operates as a lightweight distillation requiring only a standard training loop, ensuring *efficiency*.

### 4.2 PROPOSED METRICS

**MIA Adaptation for Regression.** To assess efficacy, we propose a modified MIA evaluation based on the setup proposed by Tarun et al. (2023). We maintain their enhancement strategy, where the inputs to the adversary classifier are enriched with the model's prediction, utilizing the penultimate layer gradients and activations, which provide more information than the scalar regression output alone. However, rather than training the classifier to distinguish between retain and forget samples, we focus on distinguishing between forget and test samples after unlearning, as done in many literature works (Kurmanji et al., 2023; Choi & Na, 2023). The rationale is that a successful unlearning method should make the forget data indistinguishable from unseen test data, ideally leading to a classification accuracy close to that of the gold model.

**GRUM.** To aggregate the efficacy/utility/efficiency dimensions into a single score, we adapt the GUM metric (Koudounas et al., 2025) by introducing a regression-specific variant, which we refer to as GRUM (Global Regression Unlearning Metric). We maintain the definitions of efficacy and efficiency unchanged, but modify the utility term to adopt the regression error rather than the model's classification accuracy.

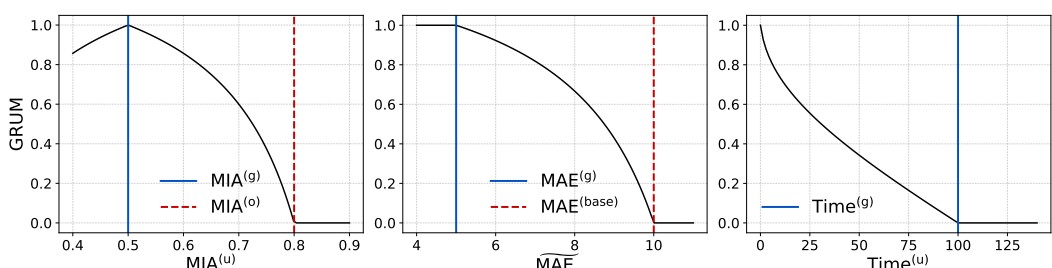

Figure 4: GRUM as a function of MIA (left), MAE (center), and unlearning time (right). Solid blue vertical lines denote the performance of the gold model, while dashed red lines indicate the MIA accuracy of the original model (left) and the baseline MAE of the naive regressor (center).

To define a meaningful utility score ($U$) in the regression setting, we compare the performance of the unlearned model to both the gold reference model and a constant baseline. Let $\text{MAE}_{\text{unl}}$ be the mean squared error of the unlearned model on the test set, $\text{MAE}_{\text{gold}}$ the MAE of a model retrained from scratch using only the retain data, and $\text{MAE}^{(\text{base})}$ is the error of a naive baseline model that always predicts the mean of the training targets.

To ensure that $U$ is bounded and interpretable, we define a clipped version of the unlearned MAE:

$$\widetilde{\text{MAE}} = \min\left(\max\left(\text{MAE}^{(\text{u})}, \text{MAE}^{(\text{r})}\right), \text{MAE}^{(\text{base})}\right) \tag{8}$$

This clipping guarantees that utility is not overestimated in cases where $\mathcal{M}_{\text{unl}}$ performs better than $\mathcal{M}_{\text{gold}}$ (which may indicate residual information from the forget set), and is also capped by the worst-case baseline. The utility component is then defined as:

$$U = \frac{\text{MAE}_{\text{const}} - \widetilde{\text{MAE}}_{\text{unl}}}{\text{MAE}_{\text{const}} - \text{MAE}_{\text{gold}} + \epsilon} \tag{9}$$

where $\epsilon$ is a small constant set for numerical stability.

This formulation ensures that $U = 1$ when the unlearned model matches the gold model, and $U = 0$ when the model performs no better than a constant predictor, as shown in Figure 4. By integrating this regression-specific utility term with efficacy (via MIA) and efficiency (via speed-up), the GRUM metric offers a unified view of unlearning quality across all three principles.

## 5 EXPERIMENTS

In this section, we first describe the experimental setup, including the datasets, model architectures, and baseline unlearning methods considered. We then present the results obtained by GURU and compare them against competing approaches across all evaluation metrics.

### 5.1 EXPERIMENT SETUP

**Datasets and Models.** To simulate realistic unlearning scenarios – such as when a user requests the deletion of their data – we construct dataset splits based on identity information. For each dataset, all samples associated with a given identity are assigned entirely to either the retain, forget, or test set, ensuring no identity overlap and preventing leakage across partitions. Formally, let each sample be $(x_i, y_i, \text{id}_i)$ with $\text{id}_i \in \mathcal{I}$ denoting its identity. We partition the identity set into three disjoint subsets $\mathcal{I} = \mathcal{I}_r \,\dot\cup\, \mathcal{I}_f \,\dot\cup\, \mathcal{I}_t$, corresponding to retain, forget, and test identities. The data splits are then induced as $\mathcal{D}_r = \{(x, y, \text{id}) : \text{id} \in \mathcal{I}_r\}$, $\mathcal{D}_f = \{(x, y, \text{id}) : \text{id} \in \mathcal{I}_f\}$, and $\mathcal{D}_t = \{(x, y, \text{id}) : \text{id} \in \mathcal{I}_t\}$, ensuring no identity overlap across partitions. The forget set is chosen so that $|\mathcal{D}_f|/|\mathcal{D}_r| + |\mathcal{D}_f| \approx 0.05$.

We conduct experiments on two computer vision (CV) and two natural language processing (NLP) datasets:

- **AgeDB** (Moschoglou et al., 2017): A curated dataset containing 16,488 face images of 568 celebrities, each annotated with the subject's age.
- **IMDB** (Rothe et al., 2015): A large-scale dataset composed of 460,723 face images collected from actor profiles on IMDB, covering 20,284 unique identities. The dataset is the IMDB subset extracted from the original IMDB-WIKI dataset.
- **IMDB Reviews** (Maas et al., 2011): This dataset contains 50,000 movie reviews labeled with sentiment scores. Each review is associated with one of 7,037 films, which we treat as identity labels for unlearning purposes. The task is to predict the sentiment score from the review text.
- **Scopus**[1]: We collected a dataset of titles, abstracts and number of citations for articles published by various authors, in 10 different areas of research, using the Scopus search engine. We collected a total of 200 authors (20 per research field), each having between 25 and 100 published articles (60 on average). We frame a regression task by predicting the number of citations received by a given article, based on their title and abstract.

For the CV datasets, we use a ResNet-18 (He et al., 2015) backbone trained for 20 epochs with a learning rate of $10^{-4}$. For the NLP ones, we fine-tuned a BERT-based (Devlin, 2018) model for 20 epochs using a learning rate of $2 \cdot 10^{-5}$.

**Baseline Methods.** We evaluate GURU against five unlearning approaches adapted to the regression setting and already proposed by Tarun et al. (2023):

- **Fine-tuning (FT):** The model continues training on the retain set only, starting from the original weights. By updating model weights baed on retain data, the influence of the forget set is reduced.
- **NegGrad (NG)** (Golatkar et al., 2020): A method that updates the model by applying the inverse of the gradient computed on the forget set. In this way, the effect of the target samples on the model's behavior is partially reversed. However, this approach often breaks the model after a limited number of updates.
- **Gaussian Amnesiac Learning (GAL)** (Tarun et al., 2023): Inspired by label randomization in classification Golatkar et al. (2020), this method replaces the true labels of the forget set with samples from a Gaussian distribution fitted to the full training label distribution. The model is then finetuned on the retain set combined with the mislabeled forget data.
- **Blindspot Unlearning (BU)** (Tarun et al., 2023): This method uses an additional model (the blindspot model), which is trained only on the retain set for a few epochs. Unlearning is performed by optimizing the original model with a composite loss that includes: (i) a standard regression loss on the retain set, (ii) an alignment term that pushes the model's outputs on the forget set to resemble those of the blindspot model, and (iii) a regularization term that minimizes the distance between intermediate activations of the two models on the forget samples.

All baselines are implemented using the same model architecture and data splits as GURU. Hyperparameters are tuned independently for each method to ensure a fair comparison and are reported in the Appendix A [2]. .

## 5.2 RESULTS

Tables 1 and 2 report results on AgeDB, IMDB, IMDB Reviews, and Scopus. For each dataset, we evaluate utility (MAE), efficacy (MIA), and efficiency (speedup over retraining), along with the unified GRUM score. It is important to note that for both MAE and MIA, an optimal outcome is one that is closest to the gold model, rather than the lowest absolute value. To perform this comparison, we also compute the gold (retrained) model.

---

[1]The public link to the HuggingFace dataset was redacted in compliance with the double-blind policy

[2]The code for our experiments is available at `https://anonymous.4open.science/r/GURU`.

Table 1: Comparison of unlearning methods on the computer vision datasets **AgeDB** and **IMDB**. Best results are shown in **bold**, and second-best are underlined. For **MAE** and **MIA**, the best results are those closest to the Retrained model.

| Method | | AgeDB | | | | | | IMDB | | | | |
|--------|------|----------|----------|-------|----------|--------|------|----------|----------|-------|----------|--------|
| | LR | $MAE_T$ | $MAE_F$ | MIA | Speedup ↑ | GRUM ↑ | LR | $MAE_T$ | $MAE_F$ | MIA | Speedup ↑ | GRUM ↑ |
| Orig. | - | 7.703 | 2.601 | 0.732 | 1.000× | 0.000 | - | 2.056 | 0.829 | 0.631 | 1.000× | 0.000 |
| Retr. | - | 7.272 | 6.773 | 0.658 | 1.000× | 0.000 | - | 2.112 | 2.114 | 0.560 | 1.000× | 0.000 |
| FT | 1e-05 | 7.380 | 2.041 | 0.730 | 12.86× | 0.074 | 1e-04 | 2.060 | 0.792 | 0.632 | 14.70× | 0.000 |
| NG | 1e-05 | 10.377 | 7.380 | 0.712 | **200.0×** | 0.456 | 1e-04 | 242.4 | 252.6 | 0.523 | **295.2×** | 0.000 |
| GAL | 1e-03 | **7.284** | 1.864 | 0.733 | 13.61× | 0.000 | 1e-05 | **2.092** | 0.848 | 0.631 | 15.35× | 0.000 |
| BU | 1e-03 | 9.618 | 9.487 | 0.607 | 7.002× | 0.387 | 1e-04 | 3.130 | **3.085** | 0.569 | 8.112× | **0.424** |
| **GURU** | 1e-04 | 7.387 | **6.639** | **0.646** | 7.662× | **0.580** | 1e-04 | 3.760 | 3.459 | **0.554** | 7.499× | 0.397 |

Table 2: Comparison of unlearning methods on the natural language process datasets **IMDB Reviews** and **Scopus**. Best results are shown in **bold**, and second-best are underlined. For **MAE** and **MIA**, the best results are those closest to the Retrained model.

| Method | | IMDB Reviews | | | | | | Scopus | | | | |
|--------|------|----------|----------|-------|----------|--------|------|----------|----------|-------|----------|--------|
| | LR | $MAE_T$ | $MAE_F$ | MIA | Speedup ↑ | GRUM ↑ | LR | $MAE_T$ | $MAE_F$ | MIA | Speedup ↑ | GRUM ↑ |
| Orig. | - | 0.942 | 0.251 | 0.697 | 1.000× | 0.000 | - | 41.00 | 10.12 | 0.817 | 1.000× | 0.000 |
| Retr. | - | 0.962 | 1.037 | 0.560 | 1.000× | 0.000 | - | 45.11 | 36.33 | 0.786 | 1.000× | 0.000 |
| FT | 1e-03 | 0.933 | 0.183 | 0.696 | 20.41× | 0.021 | 1e-05 | 40.31 | 9.990 | 0.817 | 19.57× | 0.000 |
| NG | 1e-06 | 1.102 | 0.627 | 0.690 | **414.3×** | 0.135 | 1e-05 | 74.94 | 70.04 | **0.807** | **494.0×** | 0.000 |
| GAL | 1e-03 | **0.983** | 0.272 | 0.692 | 19.34× | 0.095 | 1e-05 | 40.12 | 10.00 | 0.817 | 18.87× | 0.000 |
| BU | 1e-04 | 1.116 | **0.951** | **0.563** | 7.688× | 0.450 | 1e-04 | 74.02 | 70.51 | 0.749 | 7.786× | 0.000 |
| **GURU** | 1e-04 | 1.178 | 0.889 | 0.573 | 14.00× | **0.523** | 1e-05 | **44.16** | **14.93** | 0.809 | 13.40× | **0.371** |

On all datasets, GURU consistently achieves results comparable to the gold model across both MAE and MIA, while maintaining competitive efficiency.

It should be pointed out that some specific techniques occasionally achieves better performance w.r.t. GURU according to specific metrics: for instance, NG is consistently faster than GURU (and all other techniques). However, as is well known in the literature Choi et al. (2024), NegGrad significantly compromises the utility of the model (high MAE). Indeed, GRUM highlights this aspect: only methods that successfully balance the three unlearning dimensions achieve non-zero scores. This is highlighted, for instance, on the Scopus dataset (Table 2), NG, GAL and BU all incur in "*catastrophic forgetting*" (Jagielski et al., 2022) (as highlighted by the high MAE), whereas FT does not successfully remove *forget* samples (since it has the same MIA as the original model). Only GURU successfully balances all three aspects simultaneously.

Across other datasets, the same pattern emerges: some techniques cannot successfully balance the utility/efficacy trade-off. Instead, GURU emerges as a well-rounded approach to regression unlearning, jointly optimizing efficacy and utility while maintaining competitive efficiency.

# 6 CONCLUSIONS

We present GURU, a novel method for regression unlearning that uses residual-based Gaussians to represent predictive uncertainty and makes use of a closed-form KL divergence as the unlearning objective. This approach applies unlearning as a teacher-student distillation procedure, which has so far been applied to unlearning only in classification tasks, and not in regression. To evaluate regression unlearning in a principled manner, we further propose GRUM, a unified metric that jointly accounts for utility, efficacy, and efficiency of unlearning methods.

Extensive experiments across four different CV and NLP datasets show that GURU consistently balances the three desiderata of unlearning, both when assessed on individual metrics and when measured by the unified GRUM. Notably, some challenging datasets (e.g., Scopus) highlight how GURU is the only method that successfully achieves effective unlearning without incurring in *catastrophic forgetting*.

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

# A   HYPERPARAMETER TUNING

This appendix reports the complete learning-rate hyperparameter tuning for all unlearning methods across both computer vision (AgeDB, IMDB) and natural language processing (IMDB Reviews, Scopus) datasets. These extended tables complement the main results by showing the performance of each method under different settings. For each method and dataset, the configuration that achieves the highest GRUM is highlighted in italics. In case of ties, the configuration with the best **MIA** score is selected first, followed by the one with the best **MAE$_\text{T}$**.

Table 3: Comprehensive results of unlearning methods on the datasets **AgeDB** and **IMDB**. For each method and dataset, the best-performing learning rate configuration is reported in italics

| Method | LR | AgeDB | | | | | IMDB | | | | |
|---|---|---|---|---|---|---|---|---|---|---|---|
| | | MAE$_\text{T}$ | MAE$_\text{F}$ | MIA | Speedup ↑ | GRUM ↑ | MAE$_\text{T}$ | MAE$_\text{F}$ | MIA | Speedup ↑ | GRUM ↑ |
| Orig. | - | 7.703 | 2.601 | 0.732 | 1.000× | 0.000 | 2.056 | 0.829 | 0.631 | 1.000× | 0.000 |
| Retr. | - | 7.272 | 6.773 | 0.658 | 1.000× | 0.000 | 2.112 | 2.114 | 0.560 | 1.000× | 0.000 |
| FT | 1e-03 | 7.154 | 1.665 | 0.735 | 12.86× | 0.000 | 2.085 | 0.820 | 0.634 | 14.70× | 0.000 |
| FT | 1e-04 | 7.193 | 1.817 | 0.732 | 12.86× | 0.000 | *2.060* | *0.792* | *0.632* | *14.70×* | *0.000* |
| FT | 1e-05 | *7.380* | *2.041* | *0.730* | *12.86×* | *0.074* | 2.051 | 0.791 | 0.632 | 14.70× | 0.000 |
| FT | 1e-06 | 7.427 | 2.162 | 0.730 | 12.86× | 0.074 | 2.055 | 0.799 | 0.632 | 14.70× | 0.000 |
| NG | 1e-03 | 1872 | 1571 | 0.526 | 200.0× | 0.000 | 2478 | 2557 | 0.509 | 295.2× | 0.000 |
| NG | 1e-04 | 58.23 | 61.91 | 0.681 | 200.0× | 0.000 | *242.4* | *252.6* | *0.523* | *295.2×* | *0.000* |
| NG | 1e-05 | *10.38* | *7.380* | *0.712* | *200.0×* | *0.456* | 68.82 | 69.99 | 0.550 | 295.2× | 0.000 |
| NG | 1e-06 | 7.884 | 2.950 | 0.725 | 200.0× | 0.233 | 2.668 | 1.812 | 0.632 | 295.2× | 0.000 |
| GAL | 1e-03 | *7.284* | *1.864* | *0.733* | *13.61×* | *0.000* | 2.105 | 0.950 | 0.633 | 15.35× | 0.000 |
| GAL | 1e-04 | 7.337 | 1.944 | 0.734 | 13.61× | 0.000 | 2.067 | 0.810 | 0.632 | 15.35× | 0.000 |
| GAL | 1e-05 | 7.456 | 2.167 | 0.734 | 13.61× | 0.000 | *2.092* | *0.848* | *0.631* | *15.35×* | *0.000* |
| GAL | 1e-06 | 7.467 | 2.223 | 0.734 | 13.61× | 0.000 | 2.079 | 0.827 | 0.632 | 15.35× | 0.000 |
| BU | 1e-03 | *9.618* | *9.487* | *0.607* | *7.002×* | *0.387* | 4.118 | 4.210 | 0.539 | 8.112× | 0.374 |
| BU | 1e-04 | 7.398 | 6.030 | 0.719 | 7.002× | 0.306 | *3.130* | *3.085* | *0.569* | *8.112×* | *0.424* |
| BU | 1e-05 | 7.028 | 1.606 | 0.723 | 7.002× | 0.244 | 1.956 | 0.707 | 0.626 | 8.112× | 0.152 |
| BU | 1e-06 | 7.068 | 1.587 | 0.736 | 7.002× | 0.000 | 1.999 | 0.718 | 0.636 | 8.112× | 0.000 |
| **GURU** | 1e-03 | 10.245 | 8.893 | 0.611 | 7.662× | 0.410 | 5.128 | 5.114 | 0.534 | 7.499× | 0.286 |
| **GURU** | 1e-04 | *7.387* | *6.639* | *0.646* | *7.662×* | *0.580* | *3.760* | *3.459* | *0.554* | *7.499×* | *0.397* |
| **GURU** | 1e-05 | 7.487 | 3.176 | 0.704 | 7.662× | 0.452 | 2.068 | 0.911 | 0.626 | 7.499× | 0.150 |
| **GURU** | 1e-06 | 7.555 | 2.425 | 0.722 | 7.662× | 0.263 | 2.059 | 0.842 | 0.632 | 7.499× | 0.000 |

Table 4: Full results of unlearning methods on the datasets **IMDB Reviews** and **Scopus**. For each method, the best-performing learning rate configuration is reported in italics.

| Method | LR | IMDB Reviews | | | | | Scopus | | | | |
|---|---|---|---|---|---|---|---|---|---|---|---|
| | | MAE$_\text{T}$ | MAE$_\text{F}$ | MIA | Speedup ↑ | GRUM ↑ | MAE$_\text{T}$ | MAE$_\text{F}$ | MIA | Speedup ↑ | GRUM ↑ |
| Orig. | - | 0.942 | 0.251 | 0.697 | 1.000× | 0.000 | 41.00 | 10.12 | 0.817 | 1.000× | 0.000 |
| Retr. | - | 0.962 | 1.037 | 0.560 | 1.000× | 0.000 | 45.11 | 36.33 | 0.786 | 1.000× | 0.000 |
| FT | 1e-03 | *0.933* | *0.183* | *0.696* | *20.41×* | *0.021* | 40.58 | 9.298 | 0.821 | 19.57× | 0.000 |
| FT | 1e-04 | 0.938 | 0.201 | 0.700 | 20.41× | 0.000 | 39.46 | 10.12 | 0.821 | 19.57× | 0.000 |
| FT | 1e-05 | 0.939 | 0.230 | 0.697 | 20.41× | 0.000 | *40.31* | *9.990* | *0.817* | *19.57×* | *0.000* |
| FT | 1e-06 | 0.941 | 0.248 | 0.698 | 20.41× | 0.000 | 40.91 | 10.09 | 0.817 | 19.57× | 0.000 |
| NG | 1e-03 | 102.7 | 102.8 | 0.519 | 414.3× | 0.000 | 1652 | 1644 | 0.674 | 494.0× | 0.000 |
| NG | 1e-04 | 38.88 | 38.93 | 0.557 | 414.3× | 0.000 | 1157 | 1149 | 0.869 | 494.0× | 0.000 |
| NG | 1e-05 | 19.25 | 19.30 | 0.628 | 414.3× | 0.000 | *74.94* | *70.04* | *0.807* | *494.0×* | *0.000* |
| NG | 1e-06 | *1.102* | *0.627* | *0.690* | *414.3×* | *0.135* | 42.02 | 12.21 | 0.817 | 494.0× | 0.000 |
| GAL | 1e-03 | *0.983* | *0.272* | *0.692* | *19.34×* | *0.095* | 40.52 | 10.07 | 0.819 | 18.87× | 0.000 |
| GAL | 1e-04 | 0.980 | 0.259 | 0.698 | 19.34× | 0.000 | 38.87 | 10.62 | 0.823 | 18.87× | 0.000 |
| GAL | 1e-05 | 0.944 | 0.217 | 0.697 | 19.34× | 0.000 | *40.12* | *10.00* | *0.817* | *18.87×* | *0.000* |
| GAL | 1e-06 | 0.941 | 0.245 | 0.698 | 19.34× | 0.000 | 40.89 | 10.08 | 0.817 | 18.87× | 0.000 |
| BU | 1e-03 | 3.283 | 3.279 | 0.508 | 7.688× | 0.004 | 38.22 | 28.97 | 0.564 | 7.786× | 0.000 |
| BU | 1e-04 | *1.116* | *0.951* | *0.563* | *7.688×* | *0.450* | *74.02* | *70.51* | *0.749* | *7.786×* | *0.000* |
| BU | 1e-05 | 0.923 | 0.218 | 0.674 | 7.688× | 0.260 | 40.17 | 10.47 | 0.819 | 7.786× | 0.000 |
| BU | 1e-06 | 0.924 | 0.213 | 0.698 | 7.688× | 0.000 | 40.71 | 9.213 | 0.819 | 7.786× | 0.000 |
| **GURU** | 1e-03 | 3.286 | 3.290 | 0.481 | 14.00× | 0.000 | 119.4 | 113.4 | 0.591 | 13.40× | 0.000 |
| **GURU** | 1e-04 | *1.178* | *0.889* | *0.573* | *14.00×* | *0.523* | 83.69 | 76.60 | 0.784 | 13.40× | 0.000 |
| **GURU** | 1e-05 | 0.992 | 0.330 | 0.673 | 14.00× | 0.293 | *44.16* | *14.93* | *0.809* | *13.40×* | *0.371* |
| **GURU** | 1e-06 | 0.944 | 0.238 | 0.696 | 14.00× | 0.021 | 41.69 | 9.981 | 0.817 | 13.40× | 0.000 |

## LLM USAGE DISCLOSURE

During the preparation of this work, the authors used LLMs to correct typos and grammatical mistakes. After using this tool/service, the authors reviewed and edited the content as needed and take full responsibility for the content of the published article.

