# OpenReview forum: "GURU: Guided Unlearning via Residual Uncertainty in Regression Models"
_ICLR.cc/2026/Conference — ICLR 2026 Conference Withdrawn Submission_

### Official Review · Reviewer_gpnE · 2025-10-30

**Soundness:** 2
**Presentation:** 1
**Contribution:** 1
**Rating:** 2
**Confidence:** 5

**Summary:**

The paper tackles the machine unlearning problem for regression models, a setting less explored compared to the widely studied classification unlearning. To generalize the good teacher-bad teacher (GT-BT) paradigm to regression, the authors propose modeling the output as a Gaussian distribution with estimated mean and variance. During unlearning, the good teacher (trained on the full dataset) provides target outputs for retained samples, whereas the bad teacher (trained to forget) provides outputs for samples that should be forgotten. The student model is retrained to align with the combined outputs of GT (retain) and BT (forget), minimizing the KL divergence between the teacher and student output distributions (assumed Gaussian). To evaluate unlearning, the authors introduce a regression unlearning metric based on the normalized difference in MAE between the original and unlearned models.

**Strengths:**

- Addresses a meaningful gap in existing unlearning literature, which is dominated by classification tasks relying on logits or probabilities.
- Extending the teacher–student unlearning idea to continuous output spaces is interesting.

**Weaknesses:**

- The entire framework depends on several loosely motivated assumptions rather than principled derivations. In particular, modeling the regression output as Gaussian and re-estimating its variance via normalized absolute deviation from the ground truth lacks theoretical grounding. There is no justification for why this probabilistic form is appropriate for general regression targets, nor any analysis on how this assumption affects unlearning fidelity, variance stability, or bias in the reconstructed distribution.
- The paper does not clarify how the bad teacher is obtained (whether it is randomly initialized, trained on complementary data, or derived from the good teacher by perturbation). Since the GT-BT contrast forms the conceptual foundation of the approach, this omission makes the framework difficult to interpret or reproduce. The notion of "forgetting" is therefore not operationally well-defined.
- Instead of directly using $(\mu_r, \sigma_r^2)$ from the bad teacher, the method re-computes new distribution parameters, which appears unnecessary and may introduce instability.
- The proposed unlearning metric (difference between MAE of the original and unlearned models) is simplistic and insufficient as a primary contribution. It does not capture privacy or knowledge removal properties; it merely reflects prediction deviation. No statistical test or theoretical argument connects this MAE difference to the notion of successful unlearning.
- In (8) and (9), definitions of $\mathrm{MAE}_{\text{const}}$, $\mathrm{MAE}^{(u)}$, and $\mathrm{MAE}^{(r)}$ are unclear.

**Questions:**

see weaknesses

---

### Official Review · Reviewer_Ax3R · 2025-10-31

**Soundness:** 1
**Presentation:** 3
**Contribution:** 2
**Rating:** 2
**Confidence:** 5

**Summary:**

This paper introduces a teacher–student distillation framework for machine unlearning in regression models, an area that has received little attention compared to classification-based unlearning. The key innovation lies in modelling regression outputs as Gaussian distributions whose variance is derived from residual errors, enabling a closed-form KL divergence between the teacher’s and student’s predictive distributions. This probabilistic formulation supports both unlearning efficacy (removing forgotten data) and utility (preserving retained data). The authors also propose a regression-specific extension of GUM, an unlearning evaluation metric to evaluate efficacy, utility, and efficiency jointly. Experiments on four regression datasets demonstrate that the proposed method outperforms or matches existing baselines while maintaining efficiency.

**Strengths:**

**S1**.  Unlearning works usually evaluate on classification or generative models. This paper bridges this gap by focusing on less-studied regression problems.

**S2**. The paper is mostly written and presented clearly.

**Weaknesses:**

**W1**. The contribution of the paper is very limited. There is no clear evidence that current unlearning algorithms cannot be applied to regression problems with simple tweaks, e.g., see [1]. Also, in terms of novelty, the paper is mostly based on the existing distillation-based unlearning works and the novelty in GRUM evaluation is merely incremental.

**W2**. Figure 1 is never cited or described in the text.

**W3**. The core of the proposed method is considering the regression output values as Guassian variables. How is this different than just calculating mean square error with some Gaussian noise?

**W4**. $z$ in (6) is motivated by confidence level, but this doesn’t guarantee the statistical significance of the observed mean. This is not a proper statistical test.

**W5**. Figure 3 corresponds to three different scenarios, but I couldn’t find out what exactly these 3 cases are.

**W6**. Some of the existing unlearning methods can be applied to regression with simple tweaks. For example, SSD [2] can be easily applied to regression tasks without a major change. The experimental results without these baselines seem insufficient.

**W7**. The paper lacks sensitivity analysis on some hyperparameters, including $c$ which is considered to be the confidence interval of the output residuals.

**W8**. The paper’s unlearning scenarios are limited. For example, a study on the forget-set size can reveal how the method scales.

**Questions:**

Please see the weaknesses

---

### Official Review · Reviewer_MBBB · 2025-11-01

**Soundness:** 2
**Presentation:** 3
**Contribution:** 2
**Rating:** 4
**Confidence:** 4

**Summary:**

The paper introduces a distillation-based approach for machine unlearning in regression tasks, addressing the gap in existing literature which primarily focuses on classification. The authors propose GURU (Guided Unlearning via Residual Uncertainty), a distillation-based method that derives predictive uncertainty from residual errors by scaling residuals with a confidence factor to define Gaussian predictive distributions. Experiments on four datasets show that GURU achieves a balance across vision and language, outperforming baselines like Fine-tuning, NegGrad, Gaussian Amnesiac Learning, and Blindspot Unlearning, as measured by GRUM scores.

**Strengths:**

S1: The paper addresses an important and underexplored problem: machine unlearning for regression tasks, which has significant practical relevance for privacy-preserving machine learning.
S2:The experimental evaluation spans both computer vision and NLP domains, demonstrating some generalizability of the approach across different data modalities.

**Weaknesses:**

W1:The study's main contribution is the adaptation of teacher-student knowledge distillation to regression settings using Gaussian distributions informed by residual-based uncertainty. However, the uncertainty estimation approach (Equation 6) relies on a relatively straightforward formulation that would be strengthened by additional theoretical analysis.
W2: The paper assumes Gaussian prediction distributions without offering any empirical results. Regression residuals maybe often not Gaussian
and the authors supply no analysis to validate this critical distributional assumption.
W3:All baselines are from a single prior work (Tarun et al., 2023). No comparison with recent unlearning methods from classification that could potentially be adapted to regression (e.g., gradient-based methods, influence function approaches).
W4:No theoretical guarantees or analysis are provided. The connection between minimizing KL divergence and achieving unlearning is assumed rather than proven.

**Questions:**

D1: In Table 1 &2, there are GRUM scores of 0.000 for many methods. Does this indicate that the metric is overly stringent or poorly defined? When several methods simultaneously attain zero GRUM, the metric ceases to discriminate among them.

D2:Could you supply a formal analysis demonstrating that minimizing the KL divergence between student and teacher distributions provably induces “unlearning”? Under what conditions does this hold?

D3: The experiments focus on vision and language regression; How does GURU perform when applying it to classical regression domains like time-series forecasting where numerical values shows more complicated distributions.

---

### Official Review · Reviewer_QgA4 · 2025-11-02

**Soundness:** 3
**Presentation:** 3
**Contribution:** 3
**Rating:** 6
**Confidence:** 5

**Summary:**

This paper addresses machine unlearning for regression problems, an underexplored area relative to classification. The paper introduces GURU, a teacher–student distillation approach that (i) converts scalar regression outputs into simple probabilistic predictions via residual-derived Gaussian uncertainties, and (ii) trains a student to match a good teacher on retain data and an input-agnostic bad teacher on forget data by minimizing a closed-form KL divergence between Gaussians. The paper also proposes GRUM, a regression-aware unified metric (adapted from GUM) that aggregates efficacy (MIA), utility (MAE relative to gold and baseline), and efficiency (time). Experiments on four datasets (AgeDB, IMDB images, IMDB reviews, Scopus) compare GURU to prior regression unlearning methods and several baselines, showing that GURU achieves favorable trade-offs between forgetting, retained utility, and computational cost.

**Strengths:**

1) The paper tackles regression unlearning, a setting that presents conceptual challenges (continuous outputs) not addressed by classification-centric techniques. This is a meaningful and timely gap to fill.

2) Building upon the paper in "Deep Regression Unlearning", the paper defines predictive uncertainty from residuals and use closed-form Gaussian KL. The idea is a lightweight way to port teacher–student unlearning to regression, enabling a single analytic loss combining mean matching and uncertainty calibration.

3) Unified evaluation metric (GRUM): Extending GUM sensibly to regression and tying MAE/clipped MAE into the utility term is useful for benchmarking methods across the three desiderata. The visualization in Fig. 4 helps link the components.

4) The authors evaluate on four diverse datasets (vision and NLP regression), include multiple baselines adapted from prior work, present hyperparameter sweeps (in the appendix), and report unified GRUM scores to summarize trade-offs. Results show GURU often attains better balance between forgetting efficacy and retention utility.

**Weaknesses:**

1) GURU assumes Gaussian predictive distributions whose variance is derived directly from the absolute residual between prediction and ground truth (Eq. 6). This raises multiple points:

1-a) The procedure requires access to ground-truth labels for the training samples (Dr and Df) to compute σ. This is acceptable for the unlearning procedure (which operates on training data), but the paper should more explicitly state and justify this assumption and its implications for generalization (how well uncertainty estimation transfers to unseen test inputs where ground truth isn’t available).

1-b) The Gaussian assumption may be poor if residuals are heteroskedastic or non-Gaussian. The authors should discuss limitations, or report empirical checks (residual histograms / QQ plots) showing approximate normality on the considered datasets.

2) The bad teacher is modeled as an input-agnostic Gaussian (mean/variance summarizing retain set). This is straightforward, but possibly simplistic. It may not sufficiently represent a desired forgetting target in tasks where forgetting should not be the same distribution for all inputs (e.g., if forget identities have structured labels). The authors should discuss when an input-independent bad teacher is appropriate vs. situations where a structured bad teacher (e.g., random predictions sampled from a learned null model) would be better. Some baselines in classification unlearning use adversarial or generator-based bad teachers; a brief comparison or discussion would be helpful.

**Questions:**

1) The experiments fix the forget fraction to ≈5% (stated in Sec. 5.1). How does GURU behave for larger/smaller forget sets? Provide results (or at least discussion) for different |Df| / |D| ratios; in practice deletion requests could be single records or sizable subsets.

2) σ is obtained by |y − ŷ| / z where z = Φ⁻¹((1 + c)/2). While this is intuitive, the paper lacks sensitivity analysis showing how results vary with c (e.g., 0.9 vs 0.99) or whether calibration of σ (e.g., using train/val residual scaling) would help. An ablation on c (or equivalently z) would strengthen claims about robustness.

---

### Note · Authors · 2025-12-03

**Comment:**

I am writing to request the withdrawal of my paper. I appreciate your assistance.

**Withdrawal Confirmation:**

I have read and agree with the venue's withdrawal policy on behalf of myself and my co-authors.